# Improved Local Anesthesia at Inflamed Tissue Using the Association of Articaine and Copaiba Oil in Avocado Butter Nanostructured Lipid Carriers

**DOI:** 10.3390/ph16040546

**Published:** 2023-04-05

**Authors:** Gustavo Henrique Rodrigues da Silva, Julia Borges Paes Lemes, Gabriela Geronimo, Fabíola Vieira de Carvalho, Talita Cesarim Mendonça, Kauê Franco Malange, Fernando Freitas de Lima, Márcia Cristina Breitkreitz, Carlos Amilcar Parada, Teresa Dalla Costa, Eneida de Paula

**Affiliations:** 1Department of Biochemistry and Tissue Biology, Institute of Biology, University of Campinas—UNICAMP, Campinas 13083-862, SP, Brazil; gabrielageronimo95@gmail.com (G.G.); fabiolavieiracarvalho@hotmail.com (F.V.d.C.); talitacesarim@yahoo.com.br (T.C.M.); flfernando_@hotmail.com (F.F.d.L.); 2Department of Structural and Functional Biology, Institute of Biology, University of Campinas—UNICAMP, Campinas 13083-862, SP, Brazil; juliabplemes@gmail.com (J.B.P.L.); malangefc@gmail.com (K.F.M.); parada@unicamp.br (C.A.P.); 3Department of Analytical Chemistry, Institute of Chemistry, University of Campinas—UNICAMP, Campinas 13083-970, SP, Brazil; marciacb@unicamp.br; 4Department of Production and Control of Medicines, Faculty of Pharmacy, Federal University of Rio Grande do Sul—UFRGS, Porto Alegre 90610-000, RS, Brazil; dalla.costa@ufrgs.br

**Keywords:** drug delivery systems, nanostructured lipid carriers, inflammation, local anesthetics, articaine

## Abstract

Unsuccessful anesthesia often occurs under an inflammatory tissue environment, making dentistry treatment extremely painful and challenging. Articaine (ATC) is a local anesthetic used at high (4%) concentrations. Since nanopharmaceutical formulations may improve the pharmacokinetics and pharmacodynamics of drugs, we encapsulated ATC in nanostructured lipid carriers (NLCs) aiming to increase the anesthetic effect on the inflamed tissue. Moreover, the lipid nanoparticles were prepared with natural lipids (copaiba (*Copaifera langsdorffii*) oil and avocado (*Persia gratissima*) butter) that added functional activity to the nanosystem. NLC-CO-A particles (~217 nm) showed an amorphous lipid core structure according to DSC and XDR. In an inflammatory pain model induced by λ-carrageenan in rats, NLC-CO-A improved (30%) the anesthetic efficacy and prolonged anesthesia (3 h) in relation to free ATC. In a PGE2-induced pain model, the natural lipid formulation significantly reduced (~20%) the mechanical pain when compared to synthetic lipid NLC. Opioid receptors were involved in the detected analgesia effect since their blockage resulted in pain restoration. The pharmacokinetic evaluation of the inflamed tissue showed that NLC-CO-A decreased tissue ATC elimination rate (ke) by half and doubled ATC’s half-life. These results present NLC-CO-A as an innovative system to break the impasse of anesthesia failure in inflamed tissue by preventing ATC accelerated systemic removal by the inflammatory process and improving anesthesia by its association with copaiba oil.

## 1. Introduction

One of the major challenges to be overcome in dental practice is the failure of local anesthetics (LAs) to promote anesthesia in inflamed tissues. Indeed, in dentistry, anesthesia fails in ca. 70% of cases with inflammation and the patient is submitted to the procedure without adequate pain management [1].

LA agents used in infiltrative anesthesia—including articaine (ATC), pKa 7.8—are pH-dependent amphiphilic molecules that, in the acidic environment of inflamed tissue, are found in the protonated form with lower anesthetic potency. Moreover, the increased blood flow arising from the inflammatory process—that accelerates drug removal from the site of injection—together with nerve sensitization also contribute to anesthetic failure [2]. Even today, there is no efficient solution for inflamed tissue anesthesia. For this reason, efforts must be dedicated to developing innovative systems able to eliminate or minimize the causes that trigger anesthetic failure in the presence of inflammation. 

Nanostructured lipid carriers (NLCs) are drug delivery systems (DDS) based on a mixture of solid and liquid lipids stabilized by a surfactant [3]. NLC are excellent carriers for LAs [2], potentiating the anesthetic effect in healthy tissues. Moreover, unlike commercial injectable LA forms (pH < 6), LAs can be encapsulated in NLC formulations with pH > 8, i.e., in the neutral form (more lipid soluble and more potent). Despite this, there are no reports in the literature on the effectiveness of LA inside NLCs for pain control in inflamed tissues.

For NLC development, two kinds of lipid excipients may be used: synthetic or natural. The use of natural lipids may serve a dual purpose: structural and functional [4]. Here, we employed two natural lipids to prepare an NLC–ATC formulation, aiming to improve its anti-inflammatory and analgesic properties: avocado (*Persia gratissima*, AB) butter, AB—as the solid lipid, and copaiba (*Copaifera langsdorffii*) oil, CO—as the liquid lipid. The literature assigns the potent anti-inflammatory activity in AB to its unsaturated fatty alcohols persenone A and B [5]. Likewise, anti-inflammatory and opioid effects have been ascribed to the major component of CO, β-caryophyllene [6].

In this work, we describe the advantages of NLC for the encapsulation of ATC, an anesthetic used in the dental practice, which has better activity in inflamed tissue than any other LA [7]. NLCs were prepared only with natural lipid excipients: avocado butter and copaiba oil. The in vivo results with murine models of inflammatory pain and in situ pharmacokinetic studies proved the effectiveness of this plant-derived nanocarrier to promote anesthesia at inflamed tissue sites.

## 2. Results and Discussion

In a previous study [8], we described the preparation of different NLC formulations for the encapsulation of ATC using an experimental design (DoE) in which the selection of optimized formulations also considered their anti-inflammatory effects detected in a zebrafish model. Here, such optimized NLC formulations [8], classified by the type of their lipid excipients (natural and synthetic for comparison purposes), were employed. Their compositions (NLC-CO-A, NLC-A), and that of their controls prepared without ATC, are given in Table 1.

Table 2 shows the physical and chemical features of these optimized NLC regarding particle size (206–238 nm), polydispersity index (<0.2), zeta potential (>20 mV, in modulus), and nanoparticle concentration (>10^13^ particles/mL). Table 2 also shows the encapsulation efficiency (%EE) of ATC by these particles, which was higher than 70% in both formulations, corresponding to a loading capacity > 8%. The spherical morphology of the particles was confirmed by transmission electron microscopy as well as their ability to prolong the in vitro release of ATC [8]. Of interest, the natural ATC–nanoparticles prepared with avocado butter (AB) and copaiba oil (CO) exhibited anti-inflammatory and analgesic activity over zebrafish larvae [8].

In this work, we evaluated structural and pharmacokinetic features of the NLC–ATC formulations, along with their effects in murine models of inflammatory pain, to demonstrate the pain control applicability of these plant-derived nanotherapeutics.

### 2.1. DSC and XDR Measurements

The crystalline structure of the lipid NLC core has been assessed by gathering information obtained by differential scanning calorimetry (DSC) and X-ray diffraction (XDR) [9]. This characterization is justified since high degrees of lipid crystallinity can be associated with destabilization and drug expulsion from the core of these lipid-based nanometric systems [10].

Changes in the melting point and enthalpy of the solid lipid (SL) can be followed through DSC analyses. A decrease in the solid lipid (SL) phase transition indicates the lower crystallinity of the system [4,11]. Figure 1A shows the results obtained for the synthetic NLC formulation in which cetyl palmitate (CP) is the SL. Figure 1B shows the results related to the natural NLC formulations (NLC-CO, NLC-CO-A) where the solid lipid is AB. T80 was the nonionic detergent of both formulations. Table 3 shows the melting points and enthalpy results determined for each sample.

In Figure 1A, one can see that pure CP shows a melting peak at 57.7 °C and an enthalpy of 222 J/g. In the NLC composed of CP without ATC (NLC), a transition peak was detected at 54.8 °C (121.8 J/g). As the major component of the NLC, and the only one with a transition in this temperature range, the decrease in the transition temperature and enthalpy of CP indicates a decrease in its crystallinity while it is inside the nanoparticle. This result was observed for the SL excipients of NLC, as reported in the literature [11,12,13]. ATC melts at 70.7 °C in its basic form [8], but no endothermic event was detected at this temperature when it was encapsulated in the synthetic NLC (NLC-A). A reduced thermal event (54.2 °C; 107.8 J/g) related to CP transition was observed for this sample; when compared with NLCs without ATC (NLC = control), this can be assigned to the presence of ATC in the lipid core of the synthetic nanoparticles.

In NLCs produced with natural lipids, pure AB was used as the solid lipid, and it showed an endothermic transition at 54.7 °C with an enthalpy of 57.8 J/g (Figure 1B and Table 3). AB is a mixture of several fatty acids, whose composition varies according to the production method and place of avocado cultivation [14]. The high temperature observed for AB transition in Figure 1B suggests the presence of a large amount of saturated lipids in its composition [15]. As observed for the synthetic NLC, a decrease in the crystallinity was observed in this SL component (AB) when formulated in nanoparticles (NLC-CO). Likewise, addition of ATC further decreased the transition temperature and enthalpy (52.9 °C and 35.0 J/g, respectively) of AB, compatible with the encapsulation of the LA in the NLC-CO-A lipid core.

X-ray diffraction is another technique that can detect changes in the crystalline network of solid lipids inside lipid nanoparticles [16]. Through XRD, information about two aspects can be obtained: the formation of an amorphous structure when the SL is added to the liquid lipid to form the nanoparticle core—evidenced by a reduced intensity and/or increased peak width—and amorphization of the drug itself, which is characterized by the absence of diffraction peaks observed in its crystalline form, indicating that the drug is dispersed into the lipid matrix [13,17,18].

Figure 2A shows the diffractograms related to the synthetic NLC formulation, ATC, and CP. The diffractogram of the solid lipid shows characteristic intense peaks at 7°, 11°, 21°, and 24°, revealing the high degree of crystallinity of CP, as is reported in the literature [19]. The intensity of those peaks decreases when CP is inside the nanoparticles (NLC and NLC-A) since addition of the liquid lipid and ATC reduce the crystallinity of CP, in agreement with DSC data. The diffractogram of pure ATC (Figure 2A) shows a high intensity peak at 43° and peaks of equivalent intensities at 28°, 24°, 21°, 17°, 11°, and 10°. These peaks were not detected in the NLC-A sample, which indicates ATC encapsulation or dissolution in the lipid core of the synthetic NLC. 

Figure 2B shows the diffractogram of the natural NLC, pure AB, and ATC. Pure AB displayed intense diffraction peaks at 23°, 21°, and 6°, denoting the crystalline arrangement of the saturated AB lipids [20,21]. In NLCs, the intensities of AB peaks were significantly reduced, indicating a much less crystalline lipid core (NLC-CO and NLC-CO-A samples). As observed for the synthetic formulation, ATC peaks were not detectable in the NLC-CO-A sample, indicating LA insertion in the lipid NLC matrix.

Furthermore, according to DSC and XDR analyses, both (synthetic and natural) nanoformulations have a low crystallinity lipid core that is desirable for this kind of carrier, explaining their long-term (1 year) stability [8] and high ATC encapsulation efficiency (>70%).

### 2.2. In Vivo Tests Performed through Inflammatory Pain Models

#### 2.2.1. Subcutaneous Injection of λ-Carrageenan

λ-Carrageenan is a linear sulfated polysaccharide extracted from edible red algae that induce acute local inflammation when injected into the plantar tissue of rats. Immediately after the subcutaneous injection of carrageenan, local edema, pain, erythema, and heat are observed [22]. In this model, the development of mechanical and thermal hyperalgesia lasts for ca. 6 h and can be detected by pain behavior tests [23]. Two stages of the inflammatory process are induced by λ-carrageenan: the first—which occurs in the first hour of injection—is associated with the release of histamine, serotonin, and bradykinin; the second phase—which occurs 1–3 h or more after injection—is characterized by the release of prostaglandins [24].

Here, we measured the intensity of mechanical hyperalgesia after subcutaneous injection of λ-carrageenan by the electronic von Frey test during the entire inflammatory process (1 to 6 h after injection). The maximum intensity of pain was observed at 3 h after carrageenan injection (Figure 3B) when all inflammatory mediators were present. For this reason, the formulations to be tested were injected 10 min before the third hour (as represented in the timeline of Figure 3A).

Figure 3B shows the dose–response assessment of free ATC solution at different (0.1, 0.5, and 2%) concentrations. At 0.1% ATC, a 19% increase in hyperalgesia was observed in comparison with the control (saline-treated) group. The most likely explanation is that the acidic pH of this solution (5.2 ± 0.1) combined with the sensitivity at the application site could have generated this response. The 0.5% ATC injection did not increase the hyperalgesia (as observed with 0.1% ATC just after its application). In relation to the control group, 0.5% ATC significantly decreased (25%) the intensity of hyperalgesia measured 3 h post carrageenan injection. Finally, 2% ATC induced deep anesthesia in the animal’s paw. Therefore, this concentration (2%) was excluded from the following tests as it would not allow measuring whether the NLC-ATC formulations cause potentiating effects. 

Figure 3C shows the results obtained with all formulations containing 0.1% ATC. The NLC-A formulation (composed of synthetic lipids) reduced the mechanical hyperalgesia by 6.5% when compared with control (saline) and by 22% (*p* < 0.05) when compared with free ATC (which, at this concentration, increased the animal hyperalgesia as shown in Figure 3A). When encapsulated in the natural NLC (NLC-CO-A), 0.1% ATC promoted a much more evident effect than that observed with NLC-A, with a 34% decrease in the intensity of hyperalgesia in relation to the control. Such a decrease in the hyperalgesia for NLC-CO-A over NLC-A was also observed 1 h after application.

With 0.5% ATC (Figure 3D), shortly after application both NLC formulations showed similar results in terms of analgesia: a reduced pain sensitivity of ca. 50% when compared with control (saline) and by around 30% in relation to free ATC. In the group treated with NLC-A, the anesthetic activity was extended for 1 h after application. However, after 2 h of application, the values returned to those observed in the control group. A different profile was observed with the nanoparticles containing natural lipids (NLC-CO-A): the animals that received this treatment had an increasing improvement (decreased pain sensitivity) until the end of the experiment—hours 4 to 6.

Finally, Figure 3E shows the response to controls: injection of saline and NLC formulations without ATC. Whereas the response to NLC was similar to that of the control (saline) group, injection of the natural NLC—prepared with AB and copaiba oil (NLC-CO)—induced about 40% higher analgesia than that of the control group right after the injection. In addition, analgesia was maintained until one hour after application (hour 4 of the experiment).

The advantages of encapsulating LA into nanoparticulated DDS included, for instance, the sustained release [2] and protection of the encapsulated drug against systemic degradation [25]. In the studied ATC-in-NLC formulations, these features would be of interest in cases of inflammatory hyperalgesia, which require higher doses of LA at the injection site. Another attribute of these nanotherapeutics is their internalization by cells [26]. In inflamed tissues, where lower tissue pH increases protonation and reduces LA–membrane partitioning, internalization of the nanoparticle by cells could offer another pathway for the anesthetic agent to bind to the inactivation gate of the voltage-gated sodium channel that is located on the cytoplasmic side of the cytoplasm membrane [27]. When considered together, these attributes of the lipid-based nanoparticle systems can explain the ability of both (synthetic and natural) ATC-in-NLC formulations to increase anesthesia in inflamed tissues, as observed for the two subclinical ATC concentrations tested.

In addition, although speculative, Figure 3C,D show that free ATC (at 0.1 and 0.5%) produced an increase in pain after injection (evident at times 4–6 h post λ-carrageenan application), which could be explained by the low pH (4.8–5.2) of the commercialized (hydrochloride) ATC solution. Low pH, associated with the inflammatory process, would explain such hyperalgesia. A similar hyperalgesia effect was observed with control NLCs composed of synthetic lipids (NLC, pH~4–5) 1 h post injection. In contrast, the pH of the synthetic and natural ATC-containing NLC formulations (~8) was higher than the pKa (7.8) of ATC [2], determining the prevalence of the LA neutral form (of higher lipid partitioning), thus explaining the more effective anesthetic effect achieved with the nanoformulations.

#### 2.2.2. Subcutaneous Injection of Prostaglandin E2

Prostaglandin E2 (PGE2) is the primary pro-inflammatory prostaglandin that promotes neuronal sensitivity, one of the main signs of inflammation [28]. The PGE2 model, in contrast to the carrageenan one, does not trigger the inflammatory cascade and has no tissue acidosis; it directly activates EP receptors on the nociceptors (neuronal cells specialized in the detection and propagation of noxious stimuli), resulting in pain development. To better elucidate the effects of the NLC formulations, Figure 4 shows the results obtained after local treatment with 0.5% ATC (free and encapsulated) following PGE2 administration. As expected, free ATC significantly reduced pain intensity; however, no difference was observed in the intensity of pain (*p* > 0.05) registered after free ATC and articaine encapsulated in synthetic NLCs. Moreover, both nanoparticulated formulations (NCL-A and NCL-CO-A) reduced pain intensity by ~50% compared with the control (PGE2 plus saline). Interestingly, the formulation containing CO (NLC-CO-A) was significantly better (~20%) in terms of reduction in mechanical sensitivity than NLC prepared with synthetic lipids (NLC-A). In the natural formulation without ATC (NLC-CO), we observed a reduction in pain intensity induced by PGE2 injection. Knowing that CO shows activity in opioid receptors [25], naloxone (an opioid receptor antagonist) was co-administrated with the NLC-CO formulation. The data in Figure 4 reveal a reversal of pain reduction with NLC-CO plus Naloxone, confirming the opioid activity of CO.

#### 2.2.3. Pharmacokinetic Study (Local Microdialysis in the Inflamed Tissue)

Data from the carrageenan-induced pain model showed an increase in the anesthetic potency right after application and an extended effect achieved with ATC encapsulated either in synthetic lipids-NLC (1 h) or in NLC-OC-A (over 2 h). To better understand data obtained in the pharmacodynamic study, we decided to perform a pharmacokinetic investigation in the inflamed tissue using microdialysis to quantify unbound ATC concentrations at the site of action.

For comparison purposes, the experimental procedure was similar to that used in the study of λ-carrageenan-induced pain: after the injection of λ-carrageenan, the inflammatory process was triggered and the formulations were locally applied 10 min before the peak of the inflammatory pain (3 h after carrageenan injection (Figure 5A)). The microdialysis probe was inserted into the tissue after 1 h of carrageenan application, allowing the probe to stabilize. Figure 5B shows a photo of the probe inserted into the animal’s inflamed paw.

Figure 5C shows the results of local unbound ATC concentration measured in the inflamed paw of rats over the time after treatment with free ATC, NLC-A, or NLC-CO-A. From 35 min onwards, the concentration of unbound ATC in the group treated with free ATC was significantly lower than in the groups treated with encapsulated ATC. Additionally, the in situ LA concentration in the group treated with free ATC showed a more pronounced reduction in relation to the groups treated with NLC-A or NLC-CO-A. Therefore, these results confirm that nanoencapsulation promotes a sustained ATC release, increasing the anesthetic concentration in inflamed tissue.

The pharmacokinetic parameters determined from the local unbound ATC concentration versus time profiles (Table 4) show the influence of drug encapsulation into the nanoparticles. The ATC elimination rate constant (ke) in tissue (0.09 ± 0.01 min^−1^ when not encapsulated) decreased to 0.05 ± 0.02 min^−1^ in NLC-A and 0.04 ± 0.01 min^−1^ in NLC-CO-A. The ke values revealed a significant (*p* < 0.05) decrease in the rate of removal of the anesthetic regarding free ATC, which was systemically removed from the inflamed tissue at a faster rate. In other words, in agreement with the results of in vitro release kinetics [8], nanoencapsulation promoted a prolonged release of ATC in the inflamed tissue, maintaining higher pharmacologically active unbound ATC concentrations in situ for a longer time. Starting at 35 min from the beginning of the experiment, this effect became evident, and, probably due to this fact, we observed an effect on pain control in the groups treated with NLC-CO-A and NLC-A but not in the free ATC group. As the encapsulated fraction of ATC in NLC was high (~70%), the prolonged release allowed the maintenance of effective concentrations in the tissue for prolonged pain control.

In agreement with the ke parameter, articaine half-life (t_1/2_) in the inflamed tissue (8.4 min for free ATC) doubled (16–17 min) when the anesthetic was encapsulated in NLC-A or NLC-OC-A. Although ATC physiologically suffers degradation in the inflamed tissue, the prolonged release maintained the required ATC concentration to guarantee anesthesia for a longer time in the tissue, increasing its half-life. Of note, there are no studies in the literature reporting the pharmacokinetics of ATC in tissues, but in plasma the half-life is around 20–27 min [29,30]. In pathological peripheral inflamed tissue, the half-life of non-encapsulated (free) ATC was much shorter than that determined in plasma. However, when encapsulated, the ATC half-life resembles that reported for plasma.

The half-life and elimination rate constant data support the pharmacodynamic results showing that free ATC promoted anesthesia in less than 1 h, whereas ATC-in-NLC formulations extended the anesthetic effect for more than 1 h as it sustained the concentration of the anesthetic agent in the inflamed tissue. To our knowledge, the ability to extend the anesthetic effect has been previously reported for LA encapsulated in NLC in healthy tissues [2] but never in a model of peripheral inflammatory pain. Therefore, the prolonged release of ATC from NLC explains this remarkable effect.

One of the causes of anesthetic failure is that, due to inflammation, LA may show faster systemic absorption given the vasodilation promoted by the inflammatory process [31]. Our results demonstrate that encapsulation in NLC can neutralize this fast removal caused by tissue inflammation, keeping high levels of LA at the site of action. NLCs seem to be a promising system to avoid the adverse effect of faster drug clearance from the inflamed tissue, which could be also explored to potentiate the action of other (anti-inflammatory, analgesic, etc.) drugs.

The other pharmacokinetic parameters determined were the area under the curve (AUC) and peak concentration (C_0_) (Table 4). These did not reveal any significant difference among the groups, probably because of the large standard deviation in the data. A tendency for a higher AUC in groups treated with ATC inside NLC (~13,300–13,850 µg·min/mL) in relation to free ATC (~11,200 µg·min/mL) indicates a higher ATC unbound concentration for a longer time in groups treated with encapsulated ATC, as observed for the half-life determinations. In addition, unbound ATC peak concentration was higher (around 50%; 1149 µg/mL) in the free ATC group when compared with the NLC-A (767.1 µg/mL) and NLC-CO-A groups (610 µg/mL). This result was expected, as the free ATC fraction in the NLC-ATC samples is ~30% (%EE = 70%). Since the same ATC dose was locally administered for the three groups, the reduced unbound anesthetic peak determined for the ATC-in-NLC groups indicate that a fraction of the LA is nanoencapsulated right after dosing and that it is gradually released into the inflamed tissue, changing the local ATC elimination rate and half-life. 

The pharmacokinetic data confirm the pharmacodynamic results that show a longer anesthetic effect for ATC due to its encapsulation in the NLC, promoting a prolonged release and maintaining an effective drug level for a longer time in the inflamed tissue. Moreover, better anesthesia duration or pain reduction was detected in the group treated with natural nanoparticles (NLC-CO-A). As discussed above, the CO activity in opioid receptors is probably responsible for the better control of inflammatory pain when the natural NLCs were compared with the nanoparticles prepared with synthetic lipids. Therefore, the objective of our study was achieved with the synergism of CO and ATC.

Even in the initial stages of the test, the anesthetic effect in the groups treated with ATC inside an NLC was higher than that obtained with free ATC, even with a lower unbound concentration in the tissue (C_0_). Therefore, it is probable that other factors contributed to increase the anesthetic efficacy of these nanoformulations in inflamed tissue. One of the causes of anesthetic failure is the decrease in local pH in the inflamed tissue, which results in a decreased fraction of the neutral LA form and greater lipid solubility and membrane partitioning into the nervous membrane, where the voltage-gated sodium channel [32,33,34] is located. In this sense, the Meyer–Overton rule states that anesthetic potency increases with lipid solubility, so the neutral LA form (of higher lipophilicity) offers higher anesthetic potency [35]. In fact, ATC solution is administered in the hydrochloride form, with a pH lower than 6. The pH of the inflamed tissue—lower than the physiological pH (7.4)—also favors the protonated form of the LA to prevail, causing a lower anesthetic effect. For instance, in an inflamed tissue where the pH can be decreased by at least 0.5 units, the fraction of protonated ATC (pKa = 7.8) may be lower than 90%, explaining its low anesthetic effect.

To solve that problem, formulations with a pH that favor the existence of the neutral form of LA have been tried. Kattan et al., for instance, showed that LA administered in a buffered solution with neutral pH (7.0) promoted 2.3 times higher pulpal anesthesia than anesthetics in a hydrochloride solution [36]. Therefore, administering solutions with a higher fraction of LA in the neutral form may trigger better anesthesia. In that sense, an advantage of an NLC is that its encapsulation efficiency is higher for the neutral form of LA [2]. In the ATC-in-NLC formulations reported here (pH ~8), probably one of the factors related to the increase in the initial anesthetic activity is the presence of the neutral form of the anesthetic directly in the inflamed tissue.

The literature shows a couple of articles on the encapsulation of ATC in cyclodextrin [37], in polymeric nanocapsules [38], and in liposomes [39]. In polymeric nanocapsules and liposomes, the authors showed in vivo results of pain control using a chronic inflammation model in comparison with solutions of free ATC associated with a vasoconstrictor agent. In that case, the vasoconstrictor effect guaranteed similar anesthesia for free ATC (plus vasoconstrictor) and the nanoformulations [37]. These results show that systemic absorption of ATC can be one of the main causes of failure or decrease in anesthetic potency. In this way, the ATC-in-NLC formulations could overcome the problem of vasodilation (induced by the inflammation process) without the requirement of a vasoconstrictor agent. However, further studies should be conducted to compare the promising results obtained with the NLC-CO-A and NLC-A formulations with those of other DDS or with LA plus vasoconstrictors in a model of acute pain (which is typically the case for patients who need emergency dental treatment).

Among the formulations here proposed, NLC-CO-A shows an innovation, which is the use of natural lipids as functional excipient (with structural and biological functions). In NLC-CO-A, copaiba oil effectively controlled the inflammatory pain, with signs of improvement in the resolution of the inflammatory process since the pain was reduced after its injection until the end of the acute pain experiment (Figure 3D). However, the approval of an infiltrative formulation containing CO requires many pre-clinical studies given the diversity of compounds found in this essential oil, which may vary according to the batch and manufacturer. The main component of CO (~50%) is β-caryophyllene (BCP), a natural sesquiterpene found in several types of essential oils (including copaiba, garlic, black pepper, and cannabis sativa) [40]. BCP is a type 2-cannabinoid receptor agonist [41]. In addition to its antiedema effect [42], BCP activity in the treatment of inflammatory and neuropathic pain has already been reported [43,44]. 

Finally, we believe the next step towards the clinical application of the NLC-CO-A formulation described here would be the replacement of CO for pure BCP, a compound that is already in the market. Therefore, reproducibility and large-scale production would be guaranteed, favoring its approval for infiltrative administration in dental treatments. We anticipate that this type of innovative formulation for dental anesthesia in inflammatory pathologies with stronger anesthetic and analgesic properties could represent an important market niche in the future. Moreover, the natural plant-derived NLCs developed with AB and CO can be tested (without ATC) as coadjutants in chronic/neuropathic pain therapies. In this way, NLC can be interesting DDS carriers to improve the bioavailability of natural lipids.

## 3. Conclusions

Anesthetic failure in inflamed tissue is a major challenge in dentistry. In this study, we used nanotechnology associated with a natural excipient from the Brazilian biodiversity (copaiba oil) to solve this issue, which is intrinsic to LA. When applied to a carrageenan-induced pain model, NLC-CO-A significantly reduced the inflammatory pain and prolonged the anesthesia time in comparison with ATC in solution. In addition, the opioid activity of CO was maintained in the NLC formulation, potentiating the antinociceptive effect of ATC in inflamed tissues. Furthermore, the prevalence of the neutral ATC form in the nanoformulation was also important to guarantee anesthesia at the tissue level and under inflammatory conditions, where local acidity is one of the causes of anesthetic failure.

According to the pharmacokinetic data, NLC-CO-A provided effective levels of ATC for longer times in the inflated tissue through sustained drug release slowing down the systemic absorption of the local anesthetic. This set of results make NLC-CO-A an interesting formulation to be further explored as a dentistry LA formulation to break the impasse of anesthesia failure in inflamed tissue.

## 4. Materials and Methods

### 4.1. Materials

Articaine hydrochloride was donated by DFL Indústia e Comércio S.A. (Rio de Janeiro, RJ, Brazil); its neutral form was prepared as previously described [45]. Cetyl palmitate (CP) and Dhaykol^®^ 6040 LW (caprylic/capric triglycerides) were purchased from Dhaymers Química Fina (São Paulo, SP, Brazil). Tween^®^ 80 (T80) surfactant, λ-carrageenan (#22049), and prostaglandin E2 (#P5640) were supplied by Sigma-Aldrich (St. Louis, MO, USA). Copaiba balsam oil (CO), containing 53.62% caryophyllene according to its analysis certificate (https://www.sigmaaldrich.com/BR/pt/coa/ALDRICH/W520403/07806PI, accessed on 15 January 2023) was purchased from Sigma-Aldrich (St. Louis, MO, USA). Avocado butter, containing mainly oleic, palmitic, linoleic, palmitoleic, linolenic, and arachidonic fatty acids and less than 2% of sterols (sitosterol, campesterol), vitamins (A, B1, B2, C, and D), amino acids, and lecithin was purchased from Engenharia das Essências Ltda (São Paulo, SP, Brazil). Deionized water (18 Ω) was obtained with an Elga USF Maxima water purifier. All other reagents were of analytical grade.

### 4.2. Articaine Quantification (by HPLC) and Assessment of the Analytical Method

ATC was quantified by high-performance liquid chromatography (HPLC) with UV detection, according to the conditions described in Table 5. The conditions adopted for the analytical method were adapted from [46]. The analytical device was a Waters Breeze 2 High-Performance Liquid Chromatograph (Waters Technology., Milford, MA, USA). The limit of detection was of 0.58 µg/mL, and the limit of quantification was 1.93 µg/mL. The reproducibility was analyzed as the intra- and inter-day variability and the variability of both were less than 2%.

### 4.3. NLC Preparation

NLCs were prepared using the emulsification–ultrasonication method [47]. Both lipid and aqueous phases were briefly heated in a water bath to 10 °C above the melting temperature of the solid lipid: 67 °C for formulations with CP (melting point = 57 °C) and 65 °C for formulations prepared with AB (melting point = 55 °C). ATC, in its basic form, was then incorporated into the lipid phase. After that, the aqueous phase was added to the lipid phase drop by drop, stirring at 10,000 rpm for 3 min using a dispersing instrument (Ultra-Turrax T18, IKA WerkeStaufen, Breisgau, Germany). The resulting pre-emulsion was immediately subjected to ultrasonic agitation with a titanium microtip in the Vibracell ultrasonic processor (Sonics & Materials Incorcporation, Newtown, CT, USA) at 50 W power and a nominal frequency of 20 KHz in cycles of 30 s (on/off) for 10 min. At the end of this stage, the resulting nanoemulsion was cooled to 25 °C in an ice bath.

### 4.4. Physical and Chemical Characterization of the NLC

#### 4.4.1. Measurements of Size, Polydispersity Index, and Zeta Potential

The average particle size and the polydispersity index (PDI) were determined by dynamic light scattering (DLS) in a Zetasizer Nano ZS90 system (Malvern, UK). Measurements were performed in triplicate in polystyrene cuvettes, diluting the dispersion of nanoparticles 1000 times in deionized water at 25 °C, (10 mm path length). Data were expressed as mean ± standard deviation. The zeta potential (ZP) was determined by laser Doppler microelectrophoresis in a Doppler machine, and the measurements were performed in triplicate at 25 °C after dilution (1000x) of NLC suspensions with deionized water using appropriate polystyrene cuvettes. Data were expressed as mean ± standard deviation.

#### 4.4.2. Measurement of Particle Concentration

The concentration of particles per mL of the NLC formulations was determined with dilution (100,000x) using an NS300 NTA—Nanoparticle Tracking Analyzer (NanoSight, Amesbury, UK) equipped with a green laser at 532 nm.

#### 4.4.3. Differential Scanning Calorimetry and X-Ray Diffraction Measurements

DSC (differential scanning calorimetry) thermograms were obtained in a DSC 2910 Differential Scanning Calorimeter (TA Instruments) in a standard sealed aluminum sample holder and analyzed with Thermal Solutions v.1.25 software (TA Instruments, New Castle, DE, USA). The samples were freeze-dried (in a Freezone 4.5, Labcongo, Kansas, MO, USA) before the analyses. The heating rate was 10 °C.min^−1^ between 20 and 150 °C.

X-ray diffraction (XRD) analyses were also performed on lyophilized samples in a Shimadzu XRD7000 diffractometer (Kyoto, Japan) using a Cu-Kα source and a 2°/min run between 2θ values of 5 and 50°.

#### 4.4.4. Articaine Encapsulation Efficiency and Loading Capacity by the NLC

To evaluate the ATC content in the formulations, a sample amount of 50 µL was removed, dissolved in 1950 µL of mobile phase, and placed in an ultrasonic bath for 20 min, followed by centrifugation at 4100× *g* for 20 min. From the supernatants obtained, 200 µL aliquots were collected, diluted in 1800 µL of mobile phase, and then analyzed by HPLC as described above (item 4.2).

The encapsulation efficiency of ATC by NLC formulations was determined by centrifugal ultrafiltration [48]. Sample aliquots of 0.4 mL were transferred to a filter unit with 10 kDa pores (Millex, Millipore, Burlington, MA, USA) connected to Eppendorf tubes and centrifuged for 20 min at 4100× *g*. The filtered solution was collected, and free LA was quantified by HPLC according to the validated method (item 4.2). The percentage of encapsulation efficiency (% EE) of the anesthetic agent was calculated according to Equation (1).
(1)$$\%EE=\frac{Total\;ATC-free\;ATC}{Total\;ATC}\;\times \;100$$
where *Total ATC* corresponds to the total amount of *ATC* quantified in the NLC suspension and *free ATC* corresponds to the non-encapsulated fraction of *ATC*, quantified in the centrifugal ultrafiltration filtrate.

The amount of ATC loaded in the NLC was also expressed in terms of loading capacity (%DL—drug loading) and calculated according to Equation (2) [49]:(2)$$\%\;loading=\frac{encapsulated\;ATC\;mass}{nanoparticle\;mass}\;\times \;100$$

### 4.5. In Vivo Tests with Murine Models

#### 4.5.1. Animal Maintenance Conditions

Adult male Wistar rats (*Rattus novegicus albinus*) weighing 200–250 g were obtained from the Centro de Bioterismo at Unicamp (CEMIB/UNICAMP). They were housed in groups of 4 animals in light/dark cycles of 12 h each. All animal procedures were conducted according to the rules issued by the International Association for the Study of Pain (IASP)

#### 4.5.2. Carrageenan-Induced Inflammatory Hyperalgesia Model

The experiments were performed on 6 groups (6 rats/group). All the animals received a single intraplantar injection of carrageenan (100 µg/50 µL/paw). The mechanical nociceptive threshold was measured by electronic von Frey test at baseline conditions (before inflammatory stimulus) and 1, 2, 3, 4, 5, and 6 h after carrageenan application. The samples of saline solution, 0.1% and 0.5% ATC, 0.1% and 0.5% NLC-ATC, and control NLC without ATC (ca. 1.6 × 10^13^ nanoparticles. mL^−1^, equivalent to 0.5% NLC-ATC) were administered by intraplantar injection (50 µL/paw) 170 min after the carrageenan injection. These studies used a double-blind design. To obtain the 0.1 and 0.5% articaine concentrations, we diluted free ATC and NLC-ATC formulations before injection.

#### 4.5.3. Prostaglandin-E2-Induced Inflammatory Hyperalgesia Model

For this study, the rats were randomly separated into 8 groups of 5 animals; the mechanical nociceptive threshold was measured by the electronic von Frey test (described below) before starting any treatment (baseline threshold). This study used a double-blind design.

After measuring the baseline threshold, each animal received a single intraplantar injection of prostaglandin E2 (PGE2) at the concentration of 100 ng/50 µL/paw to induce inflammatory pain. At the pain peak—170 min after the PGE2 application—the animals received the treatment according to Table 6. Only the group treated with naloxone (20 µg/50 µL/paw) to block opioid receptors received the dose 120 min after the PGE2 application.

#### 4.5.4. Electronic von Frey Test

The mechanical nociceptive threshold was measured by the electronic von Frey test [50]. The rats were previously submitted to an adaptation in a quiet room and were placed in acrylic cages (12 × 20 × 17 cm) with a wire floor. For acclimatization, the animals remained in the cages for 15 to 30 min before the test. An angled mirror placed under the cage provided a clear view of the rear paws of the rats. The test consisted of inducing flexion of the rear paw using a polypropylene tip (0.7 mm^2^) connected to a portable force transducer (Electronic Analgesimeter, Insight^®^, São Paulo, SP, Brazil). The investigator was trained to use the tip to pinch the central region of the paw, between the five distal pads, and gradually increase the pressure. The stimulus was stopped automatically when the rat had the reflex of paw flexion. At this point, the force intensity (grams) was recorded. Three stimuli were applied to the same paw to obtain the mean response. Data were expressed as the mechanical threshold variation (Δ, in grams), which was calculated by subtracting the mean of three post-treatment measurements from the mean of pre-treatment measurements.

#### 4.5.5. Pharmacokinetic Study (Local Microdialysis in Inflamed Tissue)

The pharmacokinetic investigation of ATC in the developed formulations was performed by tissue microdialysis [51,52] using CMA 20 probes (4 mm, CMA Microdialysis, Kista, Switzerland). The probes were previously calibrated in vitro to ensure that the relative recovery by dialysis (gain) and retrodialysis (loss) were the same for ATC. The flow rate used in this study was 1.5 µL/min, maintained by a PHD22/2000 infusion pump (Harvard Apparatus, MA, USA) with a 1 mL syringe. The perfusion fluid was 0.05 M phosphate buffer, pH 7.4. The same analytical method previously described was used to quantify ATC in the microdialysate samples, which were directly injected into the HPLC system without additional processing.

For sample collection, the rats (*n* = 5/group) were previously anesthetized with urethane (1000 mg/kg). After anesthesia, the animals received a single intraplantar injection of carrageenan (100 µg/50 µL/paw). Two hours later, the probes were inserted into every animal paw. In the third hour after the carrageenan injection, formulations containing 0.5% ATC in solution or 0.5% ATC encapsulated in NLC were injected (50 µL/paw). After 3 min (to compensate for probe’s dead volume), the dialysate collection started observing 10 min intervals up to 80 min, with subsequent sample quantification by HPLC. At the end of each experiment, the buffer solution was replaced with an ATC solution at 5 µg/mL and the probe was stabilized for one hour before collecting samples for the determination of relative recovery in vivo by retrodialysis. The ATC average in vivo relative recovery was 9.4 ± 1.7%.

The real ATC unbound concentration in inflamed tissue at each sampling time was calculated using the individual probe recovery determined in vivo for each animal. Unbound ATC concentration time profiles in tissue were used to determine the individual pharmacokinetic parameters for each animal using Excel (Version 2212, Microsoft Corporation, Redmond, WA, USA). The determined pharmacokinetic parameters were: elimination rate constant (ke), calculated by the slope of the concentration-time profile in log scale; tissue half-life (t_1/2_), determined as ln2/ke; area under the curve from time zero to infinite (AUC), determined by the trapezoidal rule; and unbound peak concentration (C_0_), obtained directly from the graphs. Each parameter was statistically evaluated using ANOVA with Tukey’s multiple comparisons post hoc test in GraphPad Prism software, version 6.01 (San Diego, CA, USA).

## Figures and Tables

**Figure 1 pharmaceuticals-16-00546-f001:**
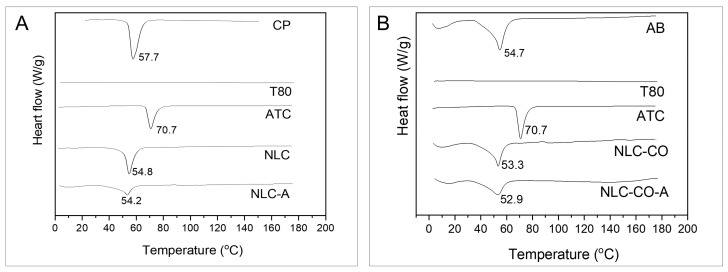
DSC thermograms obtained at a heating rate of 10 °C.min^−1^. (**A**) Pure cetyl palmitate (CP), Tween 80 (T80), articaine base (ATC), synthetic nanostructured lipid carriers without (NLC) and with ATC (NLC-A). (**B**) Pure avocado butter (AB), T80, ATC, NLC without (NLC-CO) and with ATC (NLC-CO-A). The scales of the samples are equal for comparison.

**Figure 2 pharmaceuticals-16-00546-f002:**
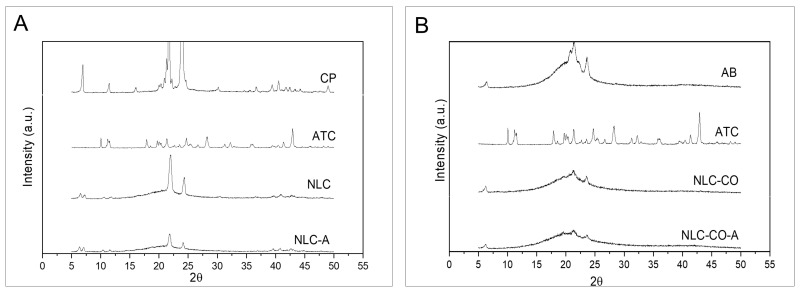
X-ray diffractograms obtained using a Cu-Kα source at a 2°. min^−1^ run. (**A**) Pure cetyl palmitate (CP), articaine base (ATC) and synthetic nanostructured lipid carriers without (NLC) and with ATC (NLC-A). (**B**) Pure avocado butter (AB), ATC and natural nanostructured lipid carriers without (NLC-CO) and with ATC (NLC-CO-A). The scales of the samples are equal for comparison purposes.

**Figure 3 pharmaceuticals-16-00546-f003:**
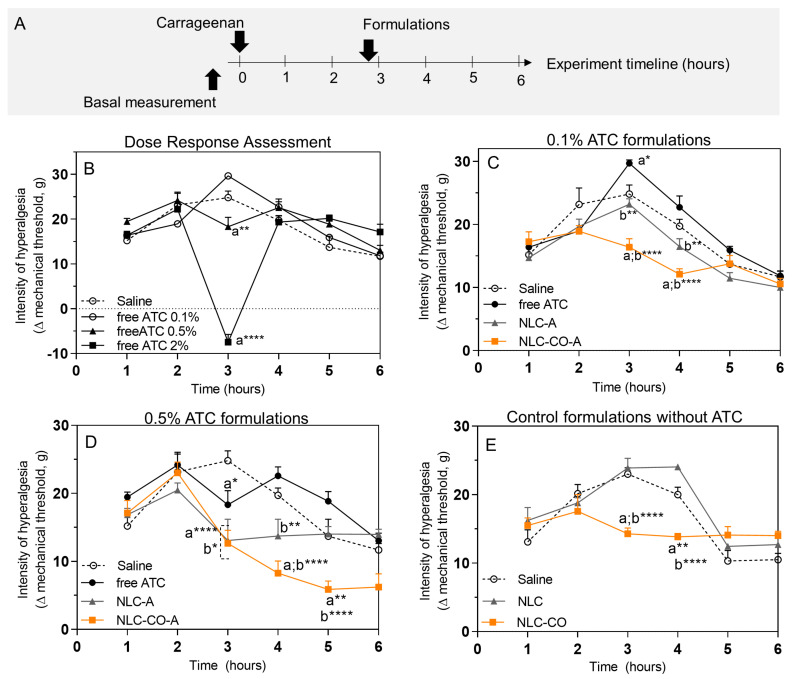
Behavior measurements of inflammatory hyperalgesia induced by λ-carrageenan in vrats. (**A**) Experimental design timeline; (**B**) mechanical nociceptive threshold measured by electronic von Frey test after treatment with saline solution (control) and free articaine solution at 0.1%, 0.5%, and 2%. (**C**) Nociceptive threshold of articaine in solution (free ATC) or encapsulated in NLC (NLC-A, NLC-CO-A) at 0.1% or (**D**) at 0.5%. (**E**) Nociceptive threshold of control NLC without articaine or prepared with synthetic lipids (NLC) or natural (NLC-CO) lipids. Statistical analysis: one-way ANOVA plus Tukey–Kramer post hoc. a, in relation to saline control; b, compared with free ATC. * *p* <0.05, ** *p* < 0.01, **** *p* < 0.001. Data were expressed by Δ (delta) of the mechanical threshold (g), calculated by subtracting the mean of three post-treatment measurements from the mean of pre-treatment measurements (baseline).

**Figure 4 pharmaceuticals-16-00546-f004:**
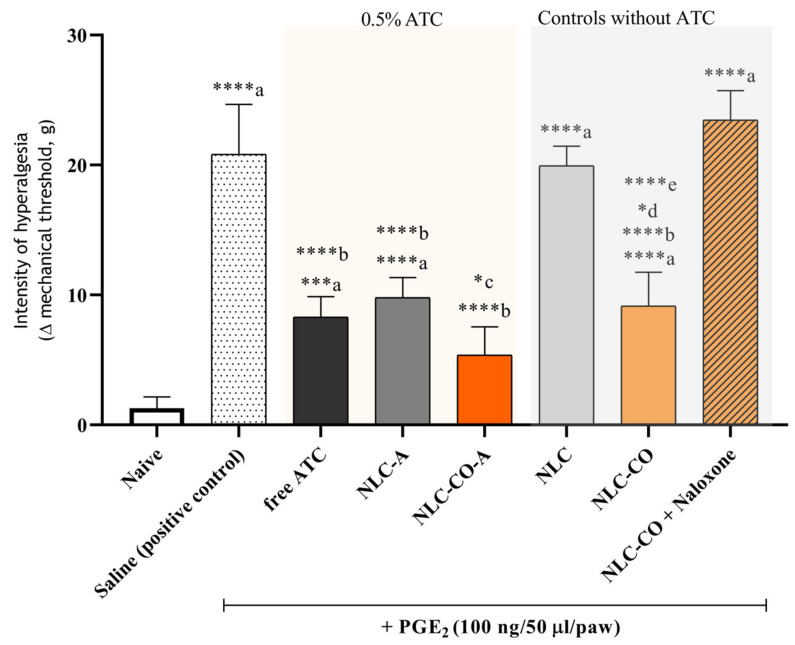
Behavior measurements of inflammatory hyperalgesia induced by prostaglandin E2 (PGE2). Statistical analysis was ANOVA with post hoc Tukey’s multiple comparisons test: a = in relation to the control (naive) animals; b = in relation to the saline group; c = in relation to the NLC-A group; d = in relation to the NLC-CO-A group; e = in relation to the NLC-CO plus naloxone group. * *p* < 0.05; *** *p* < 0.005, **** *p* < 0.001.

**Figure 5 pharmaceuticals-16-00546-f005:**
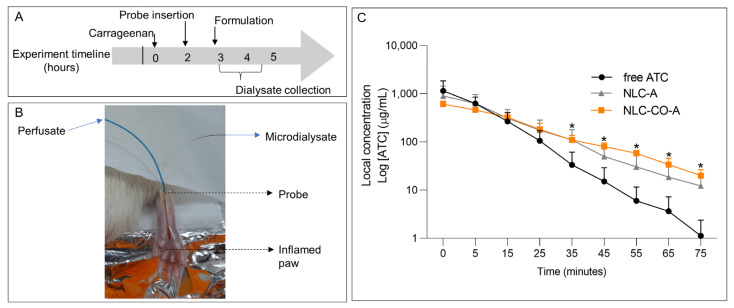
Pharmacokinetic study. (**A**) Experimental timeline. (**B**) Photo of the microdialysis probe inserted into the animal’s inflamed paw. (**C**) Local unbound articaine concentration (log [ATC]) in inflamed tissue over the time. * = statistically significant difference in relation to the free ATC group (ANOVA with post hoc Tukey’s multiple comparisons test; *p* < 0.05).

**Table 1 pharmaceuticals-16-00546-t001:** Composition of optimized synthetic and natural NLC formulations.

Lipid Source	Formulation	Solid Lipid	%	Liquid Lipid	%	Surfactant	%	Drug	%
Synthetic	NLC	Cetyl palmitate	8.75	Dhaykol^®^ *	3.75	Tween 80	3.75	-	
	NLC-A	Cetyl palmitate	8.75	Dhaykol^®^ *	3.75	Tween 80	3.75	Articaine	2
Natural	NLC-CO	Avocado butter	8.75	Copaiba oil	3.75	Tween 80	3.75	-	
	NLC-CO-A	Avocado butter	8.75	Copaiba oil	3.75	Tween 80	3.75	Articaine	2

* Dhaykol^®^ 6040 LW = caprylic/capric triglyceride.

**Table 2 pharmaceuticals-16-00546-t002:** Physicochemical properties of NLCs: particle size, polydispersity index (PDI), zeta potential (ZP), particle concentration (PC), ATC encapsulation efficiency (%EE), and drug loading (DL) capacity. Adapted from [8].

Formulation	Size (nm)	PDI	ZP (mV)	PC (^×^10^13^ Particles/mL)	%EE	DL (%)
NLC	227.1 ± 2.3	0.184 ± 0.034	−26.9 ± 0.3	7.30 ± 0.17	-	-
NLC-A	237.6 ± 3.3	0.169 ± 0.015	−42.1 ± 0.5	5.50 ± 0.40	74.0 ± 0.1	8.3 ± 0.1
NLC-CO	206.9 ± 1.0	0.161 ± 0.007	−23.7 ± 0.3	6.15 ± 0.81	-	-
NLC-CO-A	217.7 ± 0.8	0.174 ± 0.004	−40.2 ± 1.1	8.14 ± 0.71	78.4 ± 0.1	8.8 ± 0.1

**Table 3 pharmaceuticals-16-00546-t003:** Melting temperature and enthalpy determined from the DSC thermograms (Figure 1) of the NLCs and their excipients.

Sample	Melting Temperature (°C)	Enthalpy (J/g)
Pure cetyl palmitate	57.7	222.0
Pure avocado butter	54.7	57.8
Articaine	70.7	111.6
NLC	54.8	121.8
NLC-A	54.2	107.8
NLC-CO	53.3	37.6
NLC-CO-A	52.9	35.0

**Table 4 pharmaceuticals-16-00546-t004:** Pharmacokinetic parameters obtained by tissue microdialysis using free and encapsulated articaine formulations: in synthetic NLC (NLC-A) or natural NLC (NLC-CO-A).

	Formulations		
Parameters	Free ATC	NLC-A	NLC-CO-A
ke (min^−1^)	0.09 ± 0.01	0.05 ± 0.02 *	0.04 ± 0.01 *
t_1/2_ (min)	8.4 ± 1.1	15.6 ± 6.0 *	16.8 ± 4.0 *
AUC (µg·min/mL)	11,202 ± 3425	13,849 ± 6066	13,291 ± 2156
C_0_ (µg/mL)	1149.3 ± 704.5	767.1 ± 551.0	610.5 ± 95.0

* Significant difference (*p* < 0.05) in relation to free ATC (ANOVA with Tukey’s multiple comparisons post hoc test). ke = elimination rate constant, t1/2 = half-life, AUC = area under the curve, C_0_ = unbound peak concentration.

**Table 5 pharmaceuticals-16-00546-t005:** Chromatographic conditions for articaine quantification.

Parameter	Conditions
Column	C18 Gemini-NX 5µ 150 × 4.60 mm
Oven temperature	40 °C
Mobile phase	Acetonitrile:KH_2_PO_4_ 50 mM, 25:75 (*v*/*v*)
Flux	1 mL.min^−1^
Injection volume	30 µL
Wavelength	273 nm

**Table 6 pharmaceuticals-16-00546-t006:** Groups, number of animals (*n*), and treatments used in the hyperalgesia tests.

Group	*n*	Treatment
1	5	Naïve—untreated animals (control)
2	5	Saline solution (positive control)
3	5	Free articaine 0.5%
4	5	NLC-A 0.5%
5	5	NLC-CO-A 0.5%
6	5	NLC (1.6 × 10^13^ nanoparticles/mL)
7	5	NLC-CO (1.6 × 10^13^ nanoparticles/mL)
8	5	Naloxone + NLC-CO (1.6 × 10^13^ nanoparticles/mL)

## Data Availability

Data are contained within the article.

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
