# Peer review of "Improved Local Anesthesia at Inflamed Tissue Using the Association of Articaine and Copaiba Oil in Avocado Butter Nanostructured Lipid Carriers"

_pharmaceuticals, 2023, doi:10.3390/ph16040546_

Round 1
Reviewer 1 Report
Dear Authors,
In regard to your manuscript entailed "Improved anesthesia at inflamed tissue using the association of articaine and copaiba oil in avocado butter-nanostructured lipid carriers", I present herein our contribution with the peer review process. In advance, studies of dental pain management is of utmost relevance for the pharmaceutical area and, also, the manuscript had good illustrations.
- in Abstract, PK-PD would not have to be described before the abbreviation?
- line 22 seemed to need a revision to be more clear.
- please, present the botanical name of copaiba in Abstract.
- please, consider to revise the text in Introduction to improve the flow of the reading.
- all plants must be presented with the botanical name.
- Table 1 was really confusing. Was it about a published paper or from this manuscript? Reference was missing.
- please, consider re-organizing the results. It seemed out of order.
- please, consider presenting the natural lipids' compositions from where they were bought.
- please, try revising the conclusions. It was not completely focused on the objectives.
- item 4.5 described structural characterization? Please, revise this subtitle.
Reviewer 2 Report
The manuscript entitled “Improved anesthesia at inflamed tissue using the association of articaine and copaiba oil in avocado butter-nanostructured lipid carriers” deals with the evaluation of lipid nanocarriers (NLC) for the topical delivery of the anesthetic articaine (ATC).
Although the topic is interesting, many issues should be addressed as the experimental protocol is not well developed and the results are not properly presented.
Line 21. The meaning of the abbreviation PK-PD should be explained.
Line 160. Tween 80 (T80) is not shown in Fig. 2, therefore Tween 80 (T80) should be deleted.
Line 175. The authors performed in vivo tests using NLC loading 0.1% or 0.5% ATC but they did not report any data about the characterization of such NLC. On the contrary, the authors reported the characterization of NLC loading 2 % ATC that were not used in in vivo tests. The authors should report the characterization (mean size, polydispersity index, zeta potential, morphology, in vitro release kinetics) of the NLC they used to carry out in vivo tests and they should remove the data about NLC loading 2% ATC.
Line 208. The authors claim, “0.5% ATC injection did not increase the pain (as 0.1%)” but at line 205 the authors state, “At 0.1% ATC, a 19% hyperalgesia increase was observed”. Therefore, it is unclear if 0.1% ATC increased pain. Please, explain this discrepancy.
Line 248. The statement “Figures 3C and 3D show that free ATC – at 0.1 and 0.5%– produced an increase in pain after injection” contradicts the assertion at line 208 (0.5% ATC injection did not increase the pain). Please, explain this discrepancy.
Line 253. The assertion “the pH of the synthetic and natural ATC-containing NLC formulations (8-8.5) was higher than that of ATC pKa (7.8) [2] determining the prevalence of the LA neutral form - of higher lipid partitioning - thus explaining its more effective anesthetic effect” is questionable. As ATC was incorporated into NLC (therefore ATC was inside the nanocarrier), how could the pH of the medium affect ATC presence as neutral form? Please, explain.
Line 479. The authors should report limit of detection, limit of quantification, intra- and inter-day variability of the HPLC method.
Line 546. The authors lyophilized NLC samples prior to performing DC and X-ray diffraction analyses. How did the authors lyophilize NLC samples and why?
Line 556. The approval of the in vivo tests by the Ethics Committee (protocol code and date of approval) must be reported.
Line 560. Carrageenan-induced inflammatory hyperalgesia experiments were performed on six rats. How could the authors get data with standard deviation using only one animal for each formulations? Please, explain.
Line 564. What was the volume of sample injected in the rat paw?
Round 2
Reviewer 2 Report
The authors revised the manuscript properly.